# Efficient Deep Learning with Decorrelated Backpropagation

## Abstract

The backpropagation algorithm remains the dominant and most successful method for training deep neural networks (DNNs). At the same time, training DNNs at scale comes at a significant computational cost and therefore a high carbon footprint. Converging evidence suggests that input decorrelation may speed up deep learning. However, to date, this has not yet translated into substantial improvements in training efficiency in large-scale DNNs. This is mainly caused by the challenge of enforcing fast and stable network-wide decorrelation. Here, we show for the first time that much more efficient training of very deep neural networks using decorrelated backpropagation is feasible. To achieve this goal we made use of a novel algorithm which induces network-wide input decorrelation using minimal computational overhead. By combining this algorithm with careful optimizations, we achieve a more than two-fold speed-up and higher test accuracy compared to backpropagation when training a 18-layer deep residual network. This demonstrates that decorrelation provides exciting prospects for efficient deep learning at scale.

## 1 Introduction

Modern AI relies heavily on deep learning (DL), which refers to the training of very deep neural network (DNN) models using massive datasets deployed on high-performance compute clusters (LeCun et al., 2015). The established way of implementing learning in artificial neural networks is through the backpropagation (BP) algorithm (Linnainmaa, 1970; Werbos, 1974). BP is a gradient-based method that implements reverse-mode automatic differentiation to compute the gradients needed for parameter updating in neural networks (Baydin et al., 2018).

At the same time, training of DNNs consisting of many layers of nonlinear transformations is computationally expensive and has been associated with significant energy consumption at a global scale (García-Martín et al., 2019; de Vries, 2023; Thompson et al., 2021; Luccioni et al., 2023b; Strubell et al., 2019; Luccioni et al., 2023a). Hence, we are in need of green AI solutions which can significantly reduce the energy consumption of modern AI systems (Gibney, 2022; Schwartz et al., 2020; Wang, 2021). Patterson et al. (2022) describe several ways to reduce the carbon consumption of deep learning, focusing mostly on more effective deployment methods. Another key reason for deep learning's large carbon consumption, however, is the inefficiency of the backpropagation algorithm itself. BP requires many iterations of gradient descent steps until parameters converge to their optimal values. For example, training a large GPT model can easily take several weeks on a large compute cluster (Brown et al., 2020).

In this paper, we show that deep learning can be made much more efficient by enforcing decorrelated inputs throughout the network. Decorrelation has previously been proposed to make credit assignment more efficient in neural networks (LeCun et al., 2012). Intuitively, if a network layer's inputs have highly correlated features, it will be more difficult for the learning algorithm to perform credit assignment as it is now unclear if a change in the loss should be attributed to feature $i$ or feature $j$ in case both are correlated. Previous work has indeed shown that promoting decorrelation positively impacts training in relatively shallow networks (Ahmad et al., 2023). This advantage of decorrelation can be theoretically understood as a way to align the gradient update with that of the natural gradient (Ahmad, 2024; Amari, 1998).

Similar results have been obtained when inputs are whitened such that inputs are not only decorrelated but also forced to have unit variance (Luo, 2017). In (Huangi et al., 2018), whitening has been shown to have

a positive impact in training deep networks, resulting in faster convergence when measured by number of epochs and somewhat lower test error compared to regular BP. However, due to computational overhead this did not yet translate into a substantial reduction in wall-clock training time. Desjardins et al. (2015) also demonstrate that whitening has benefits when training DNNs, though this did not result in faster convergence to the minimal validation loss compared to standard BP using batch normalisation. At the same time, (Wadia et al., 2021) suggest that strict whitening can have a negative impact on generalization performance.

Hence, while the potential benefits of decorrelation and whitening are encouraging, their application for significantly reducing real-world training times in modern deep neural networks remains to be shown. This has two main reasons. First, enforcing network decorrelation during training is associated with significant computational overhead, thereby eliminating any gains in training efficiency. Second, it is not obvious how to effectively implement decorrelation in a stable manner in deep (convolutional) neural networks.

In this paper we demonstrate, for the first time, that training of large-scale DNNs can be made much more efficient through the development of a novel decorrelated backpropagation (DBP) algorithm. DBP combines automatic differentiation with an efficient iterative local learning rule which effectively decorrelates layer inputs across the network. This iterative decorrelation procedure was previously introduced by Ahmad et al. (2023) and has been shown to speed up learning in limited-depth fully-connected neural networks. Here, we extend the procedure to more efficiently and effectively train deep residual networks (He et al., 2016). This is achieved by making the procedure suitable for application in convolutional layers and ensuring that decorrelation remains stable across layers. Furthermore, we show how the same algorithm can be used to whiten layer inputs, allowing us to compare the efficacy of decorrelation and whitening in DNNs.

In the following, we show that DBP yields a two-fold reduction in training time, while achieving better performance compared to regular BP. Hence, widespread application of our approach can yield a substantial reduction in the carbon consumption of modern deep learning.

## 2 Methods

### 2.1 Decorrelated backpropagation

Let us consider a DNN consisting of $K$ layers implementing some parameterized nonlinear transform. Deep learning typically proceeds via reverse-mode automatic differentiation, also known as the backpropagation (BP) algorithm. Backpropagation updates the parameters $\boldsymbol{\theta}$ (weights and biases) of a network according to

$$\boldsymbol{\theta} \leftarrow \boldsymbol{\theta} - \eta \nabla_{\boldsymbol{\theta}} \mathcal{L} \qquad (1)$$

where $\eta$ is the learning rate, $\boldsymbol{\theta}$ are the network parameters and $\nabla_{\boldsymbol{\theta}} \mathcal{L}$ is the gradient of the loss.

Decorrelated backpropagation (DBP) proceeds as regular backpropagation but additionally enforces decorrelated inputs to layers in the network. Since decorrelation is imposed independently in each layer, we concentrate on describing the decorrelation procedure for one such layer, implementing a nonlinear transformation

$$\mathbf{y} = f(\mathbf{W}\mathbf{x}) \qquad (2)$$

with $\mathbf{W} \in \mathbb{R}^{d \times d}$ the weights and $\mathbf{x} \in \mathbb{R}^d$ the input. This transformation may be either the kernel function of a convolutional layer applied to a (flattened) input patch or the transformation in a fully-connected layer. Note further that we ignore biases without loss of generality since these can be represented using weights with fixed constant input. Input decorrelation refers to the property that $\langle \mathbf{x}\mathbf{x}^\top \rangle$ is diagonal for inputs $\mathbf{x}$, where the expectation is taken over the data distribution. In case of whitened inputs, $\langle \mathbf{x}\mathbf{x}^\top \rangle$ is assumed to be the identity matrix (Kessy et al., 2015). We will refer to the former as the covariance constraint and the latter as the variance constraint.

To ensure that the input $\mathbf{x}$ is decorrelated, we assume that it is the result of a linear transform

$$\mathbf{x} = \mathbf{R}\mathbf{z} \qquad (3)$$

where the decorrelating matrix $\mathbf{R}$ transforms the correlated input $\mathbf{z}$ into a decorrelated input $\mathbf{x}$. Hence, the transformation in a layer is described by $\mathbf{y} = f(\mathbf{W}\mathbf{R}\mathbf{z}) = f(\mathbf{W}\mathbf{x}) = f(\mathbf{A}\mathbf{z})$ with $\mathbf{A} = \mathbf{W}\mathbf{R}$. The output $\mathbf{y}_l$ of the $l$-th layer becomes the correlated input $\mathbf{z}_{l+1}$ to the next layer.

## 2.2 Decorrelation learning rule

To implement decorrelated backpropagation, we need to update individual decorrelation matrices for each of the layers in the network. This can be achieved by minimizing $K$ decorrelation loss functions *in parallel* to minimizing the BP loss with the aim of accelerating learning speed.

One way to achieve whitening as a restricted form of decorrelation is via established approaches such as zero-phase component analysis (ZCA) (Bell & Sejnowski, 1995; 1997). However, naive application of ZCA is prohibitively costly in practice since it requires application of the algorithm to the inputs of each layer for all of the examples in the dataset at each step of gradient descent. More efficient batch implementations of whitening have been shown to have a positive impact on neural network training (Luo, 2017; Desjardins et al., 2015; Huangi et al., 2018) but this did not yet yield a substantial reduction in training time in large-scale models when considering the moment at which minimal validation loss is achieved. See Appendix A for a complexity analysis of the different approaches.

To enable efficient decorrelation and whitening, we extend a recently introduced approach that enables more efficient network-wide decorrelation (Ahmad et al., 2023) and make this suitable for application in large-scale models. To derive our decorrelation learning rule, we start by defining the total decorrelation loss $\langle L \rangle$ as the average decorrelation loss across layers. The layer-specific decorrelation loss is defined as $L = \langle \ell(\mathbf{x}) \rangle$, where the expectation is taken over the empirical distribution and $\ell(\mathbf{x}) = \sum_{i=1}^{d} \ell_i(\mathbf{x})$ is the sum over unit-wise losses $\ell_i$. We define the unit-wise loss as

$$\ell_i(\mathbf{x}) = (1 - \kappa)\frac{1}{2}\sum_{j \neq i}(x_i x_j)^2 + \kappa\frac{1}{4}(x_i^2 - 1)^2 \tag{4}$$

where the regularization parameter $\kappa$ interpolates between the covariance constraint ($\kappa = 0$) and the unit variance constraint ($\kappa = 1$) such that that $\kappa \in (0, 1)$ promotes whitening. It follows that we may write the layer-wise decorrelation loss compactly as $\ell(x) = \langle (1 - \kappa)\text{Tr}(\mathbf{C}\mathbf{C}^\top) + \kappa\text{Tr}(\mathbf{V}\mathbf{V}^\top) \rangle$ where $\mathbf{C} = \mathbf{x}\mathbf{x}^\top - \mathbf{D}$ with $\mathbf{D} = \text{diag}(x_1^2, \ldots, x_d^2)$ and $\mathbf{V} = \text{diag}(x_1^2 - 1, \ldots, x_d^2 - 1)$.

Since the decorrelation loss is minimized independently in each layer, we concentrate on describing the minimization of the decorrelation loss for one such layer, as in Eq. 2. Let us first consider decreasing the correlation between variables for one input vector $\mathbf{x}$. This can be achieved using an update step of the form

$$\mathbf{x} \leftarrow \mathbf{x} - \epsilon\nabla_\mathbf{x}\ell \tag{5}$$

with $\epsilon$ the decorrelation learning rate. To compute the gradient, we first consider the partial derivative of the unit-wise loss with respect to the input $x_i$ given by

$$\frac{\partial \ell_i}{\partial x_i} = (1 - \kappa)\sum_{j\,:\,j \neq i}(x_i x_j)x_j + \kappa(x_i^2 - 1)x_i \,. \tag{6}$$

If we vectorize across units, we obtain $\nabla_\mathbf{x}\ell = (1 - \kappa)\mathbf{C}\mathbf{x} + \kappa\mathbf{V}\mathbf{x}$. If we plug this into the update step in Eq. 5, we obtain $\mathbf{x} \leftarrow \mathbf{x} - \epsilon\left((1 - \kappa)\mathbf{C} + \kappa\mathbf{V}\right)\mathbf{x}$. Ultimately, we aim to derive an update rule for $\mathbf{R}$ instead of $\mathbf{x}$. This can be obtained using the identity $\mathbf{x} = \mathbf{R}\mathbf{z}$ since this allows us to write $\mathbf{R}\mathbf{z} \leftarrow \mathbf{R}\mathbf{z} - \epsilon\left((1 - \kappa)\mathbf{C} + \kappa\mathbf{V}\right)\mathbf{R}\mathbf{z}$. Dividing by $\mathbf{z}$ on both sides and averaging over input samples we obtain our decorrelation learning rule

$$\mathbf{R} \leftarrow \mathbf{R} - \epsilon\left\langle(1 - \kappa)\mathbf{C} + \kappa\mathbf{V}\right\rangle\mathbf{R} \,. \tag{7}$$

This decorrelation rule allows for efficient batch updating of $\mathbf{R}$ since it only requires computing the decorrelated input covariance followed by multiplication with the decorrelation matrix. Note that for $\kappa = 0$ we only decorrelate the input and for $\kappa = 1$ we only enforce the unit variance constraint. For $0 < \kappa < 1$ we include both constraints to induce input whitening. Figure 1 shows effective decorrelation and whitening on two-dimensional input data.

## 2.3 Application to deep residual networks

Effective training of deep neural networks using DBP requires several modifications that improve algorithm stability, learning speed and allow application to convolutional rather than fully connected layers. As a

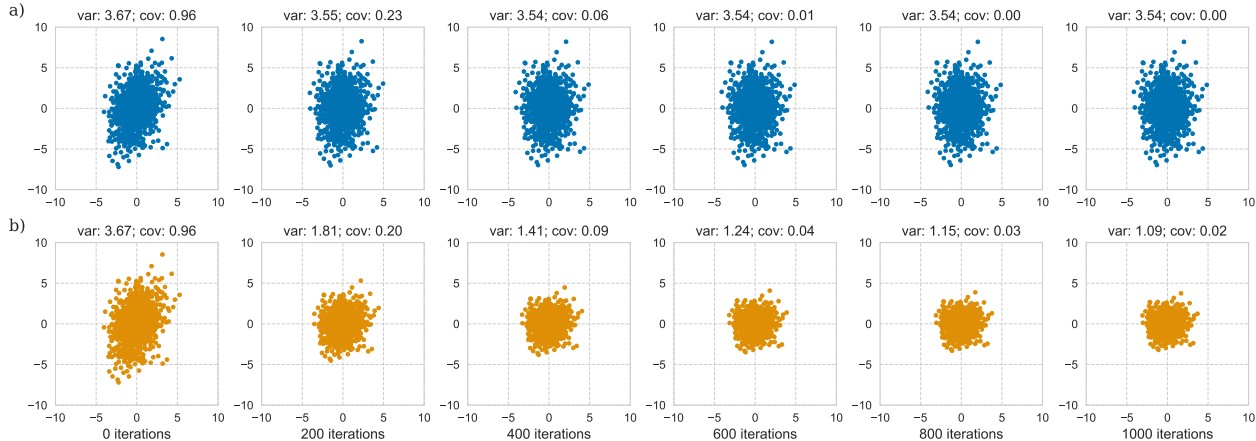

Figure 1: Demonstration of the decorrelation rule on correlated input data consisting of 1000 examples and two covariates with decorrelation learning rate $\epsilon = 0.001$. a) Decorrelation using $\kappa = 0$. b) Whitening using $\kappa = 0.5$. Mean variance and covariance reported for different iterations.

canonical example, we consider an 18-layer deep residual network (ResNet). A ResNet consists of multiple residual blocks that implement modern network components such as convolutional layers and residual connections (He et al., 2016). In the following, we describe the modifications that are required for effectively applying DBP in deep neural networks.

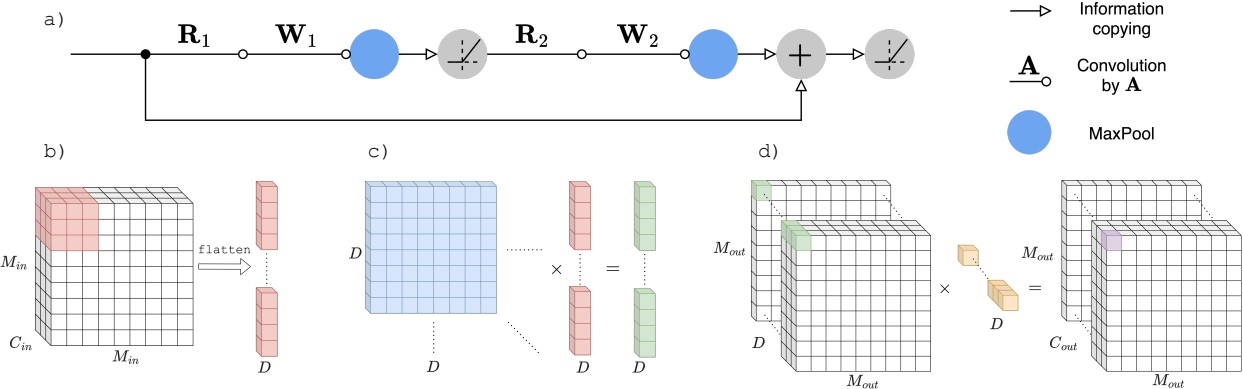

Figure 2: Implementation of decorrelated backpropagation in residual networks. a) Residual blocks as implemented by our networks. b) Patch-wise flattening of the inputs with a flattened dimension of $d = M \times M \times C_{in}$. c) Decorrelating/whitening transform of the data by decorrelation matrix $\mathbf{R}$. d) $1 \times 1$ convolution operation with the weights $\mathbf{W}$ on the decorrelated input patches.

First, we apply the decorrelation learning rule to convolutional layers as follows. Consider a convolutional layer with input size $S = M_{in} \times M_{in} \times C_{in}$, where $M_{in} \times M_{in}$ is the size of the feature map and $C_{in}$ is the channel dimension. Naive implementation of decorrelation would require updating a $S \times S$ decorrelation matrix $\mathbf{R}$, which is computationally too prohibitive in case of large feature maps. Instead, as shown in Fig. 2, we do not decorrelate the layer's entire input, but only the local image patches, so that for an image patch with dimensionality $D = K \times K \times C_{in}$, matrix $\mathbf{R}$ would be of much smaller size $D \times D$. The output of this patchwise decorrelation operation would then be $D$ for every image patch, after which we can apply a $1 \times 1$ convolutional kernel of size $D \times C_{out}$ for the forward pass. This local decorrelation is sufficient to ensure more

efficient learning of the kernel weights. Figure 2 depicts the structure of a residual block that is extended to implement decorrelating transforms.

Second, applying the decorrelation learning rule can become costly when averaging over input samples in Eq. 7 in case of convolutional layers. This is due to the need to compute an outer product $\mathbf{XX}^\top$ with $\mathbf{X} \in \mathbb{R}^{D \times p}$ the input to a layer, where $p$ is the product of the number of batch elements times the number of patches. We found empirically that updates of $\mathbf{R}$ need not use the entire mini-batch to learn the correlational structure of the data. Sampling just 10% of the samples in each batch yields almost identical performance while significantly reducing the computational overhead of computing $\mathbf{R}$'s updates, as shown in Appendix B

Third, another optimization that decreases the computational overhead of the decorrelating transform is to combine the matrices $\mathbf{W}$ and $\mathbf{R}$ into a condensed matrix $\mathbf{A} = \mathbf{WR}$ prior to multiplying by the correlated input $\mathbf{z}$. The dimensionality of $\mathbf{z}$ is much larger than that of $\mathbf{W}$ due to the batch dimension, and thus we replace an expensive multiplication $\mathbf{W}(\mathbf{Rz})$ by a cheaper multiplication $(\mathbf{WR})\mathbf{z}$, significantly reducing the time required for performing a forward pass. As an additional benefit, when training is complete, we only need to store the $\mathbf{A}$ matrices.

### 2.4 Experimental validation

To evaluate learning performance, an 18-layer deep ResNet model was trained and tested on the ImageNet Large Scale Visual Recognition Challenge (ILSVRC) dataset (Russakovsky et al., 2015). This dataset spans 1000 object classes and contains 1,281,167 training images, 50,000 validation images and 100,000 test images (Russakovsky et al., 2015).

Data preprocessing consisted of the following steps. First, images were normalized by subtracting the means from the RGB-channels' values and dividing by their standard deviation. Next, images were rescaled to $256 \times 256$ and a $224 \times 224$ crop was taken from the center. We used no data augmentation, but shuffled the order of the data every epoch.

All models and algorithms were implemented in PyTorch (Paszke et al., 2019) and run on a compute cluster using Nvidia A100 GPUs and Intel Xeon Platinum 8360Y processors. To initialize the weights of our models, we set $\mathbf{R}$ to the identity matrix and used He initialization (He et al., 2015) for $\mathbf{W}$. Models were trained for 50 epochs to minimize the categorical cross-entropy loss. Exploratory analysis revealed negligible performance difference as a function of batch size and a batch size of 256 was chosen. Instead of using the full batch of images in each decorrelation update step, we reduce the number of samples to 10%. We empirically found this value to show negligible loss in performance, while significantly speeding up runtime. The accuracy reported is the top-1 performance of the models. Reported wall-clock time was measured as the runtime of the training process, excluding performance testing.

Performance of DBP using either decorrelation ($\kappa = 0$) or whitening ($\kappa = 0.5$) was compared against regular backpropagation. For updating $\mathbf{W}$, we use the Adam optimizer (Kingma & Ba, 2014) with a learning rate of $\eta = 1.6 \cdot 10^{-4}$ and optimizer parameters $\beta_1 = 0.9$ and $\beta_2 = 0.999$. We added a small value of $10^{-8}$ to the denominator of the updates for numerical stability. The decorrelation matrix $\mathbf{R}$ is updated with stochastic gradient descent using a learning rate of $\epsilon = 10^{-5}$. To ensure a fair comparison between algorithms, we made sure that all algorithms attained their highest convergence speed by performing a two-dimensional grid search over the BP ($\eta$) and decorrelation ($\epsilon$) learning rates. See Appendix C for the grid-search results.

## 3 Results

In the following, we analyse the performance of BP versus DBP on the 18-layer ResNet model trained on the ImageNet task.

### 3.1 DBP effectively decorrelates inputs to all network layers

In Fig. 3, the change is input correlation is shown for each layer during the training process for DBP. To quantify input correlation we use the mean squared values of the strictly lower triangular part of the $\mathbf{C}$ matrix.

It can be seen that layer inputs become rapidly decorrelated during the first 10 epochs and correlations remain consistently low. This demonstrates that the decorrelating learning rule indeed effectively learns to decorrelate the inputs to all network layers.

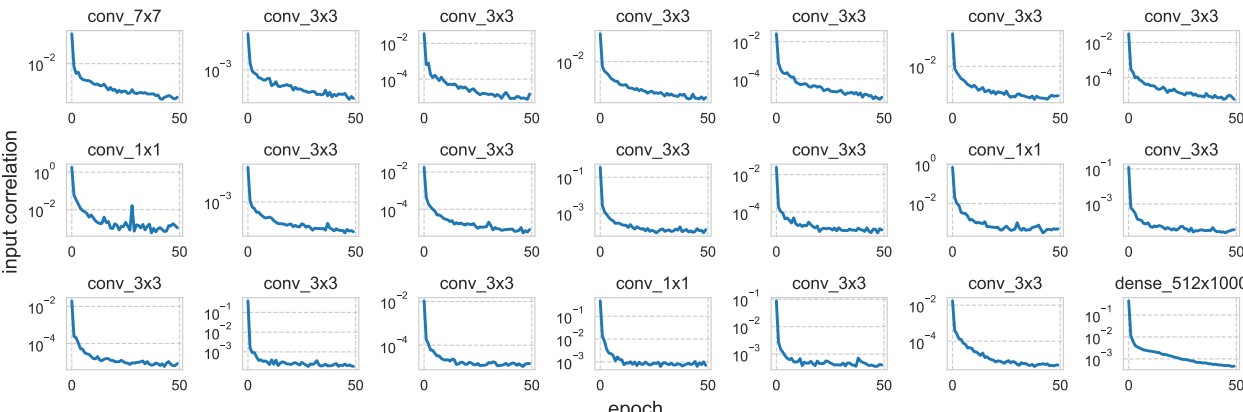

Figure 3: Input decorrelation when training a ResNet-18 model on ImageNet for 50 epochs. Network layers are ordered from left to right and from top to bottom. Panel titles indicate the layer type.

## 3.2 DBP converges much faster than BP

We observe a significant improvement when training the ResNet model using DBP compared to regular BP. As shown in Fig. 4, DBP's train accuracy (Fig. 4a) improves much faster than that of BP, indicating more effective training (computational work per epoch). The same observation holds for test accuracy (Fig. 4b). Additionally, DBP's test accuracy (55.2%) also peaks above BP's test accuracy (54.1%), while doing so in roughly half the number of epochs. Finally, we observe that a slightly higher test accuracy is achieved for whitening ($\kappa = 0.5$) compared to decorrelation ($\kappa = 0$), demonstrating the added benefit of whitening for this task.

The same patterns can be observed for the loss curves in Appendix D. Furthermore, as shown in Appendix E, a slightly lower test loss at equal train loss is obtained for DBP compared to BP, which shows that this does not come at the expense of generalization performance.

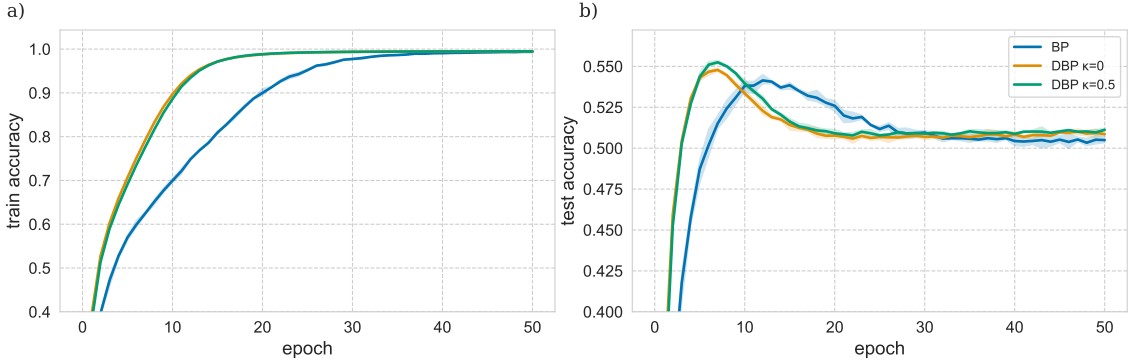

Figure 4: Performance of BP and DBP using $\kappa = 0$ (decorrelation) and $\kappa = 0.5$ (whitening). Reported results are the average of three randomly initialized networks, where minimal and maximal value are indicated by the error bars. a) Train accuracy as a function of the number of epochs. b) Test accuracy as a function of the number of epochs.

### 3.3 DBP training can be achieved at shorter wall-clock times than BP

Recall that DBP requires additional computation in each epoch due to application of the decorrelation update rule in Eq. 7. This implies that faster convergence as a function of the number of epochs may not necessarily translate into more efficient training. Therefore, we also compare wall-clock times in Fig. 5.

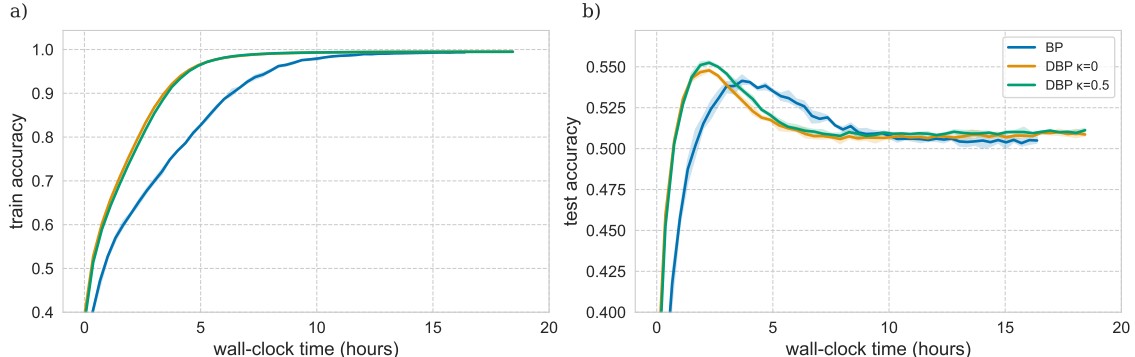

Figure 5: Performance of BP and DBP using $\kappa = 0$ (decorrelation) and $\kappa = 0.5$ (whitening). Reported results are the average of three randomly initialized networks, where minimal and maximal value are indicated by the error bars. a) Train accuracy as a function of wall-clock time. b) Test accuracy as a function of wall-clock time.

Even though DBP takes 13% more time per epoch, it still massively improves training efficiency over BP in terms of wall-clock time since it reaches maximal test performance much more rapidly than BP, taking only 2.3 hours instead of 3.7 hours, offsetting the additional compute needed. The same patterns can be observed for the loss curves in Appendix D. An overview of peak performance levels and their associated convergence speed is provided in Appendix F.

### 3.4 DBP significantly reduces carbon emission

Figure 6 visualizes what the efficiency gains are when using DBP ($\kappa = 0.5$) over BP in terms of carbon emission.

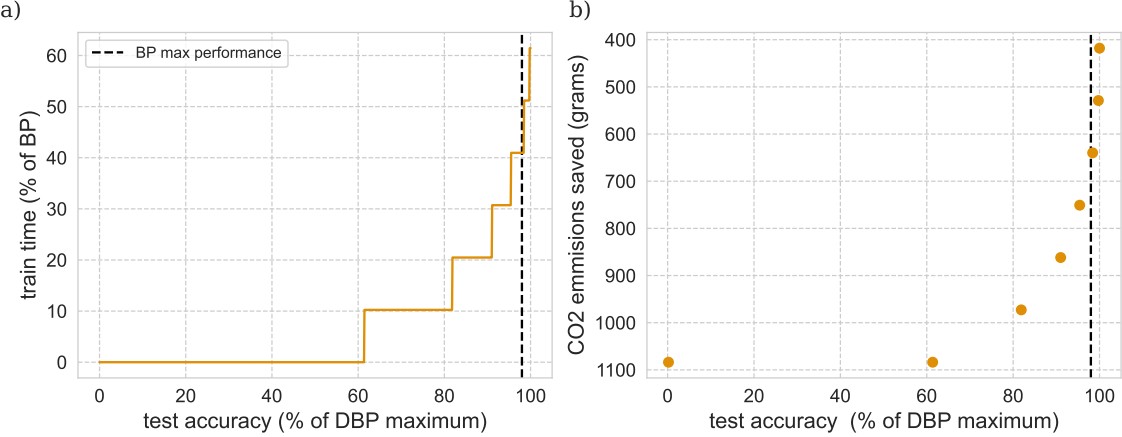

Figure 6: Efficiency gains when using DBP over BP. a) Training time required relative to BP's time to maximal accuracy to reach a specific percentage of the maximum performance of DBP. b) Estimated reduction in carbon consumption when using DBP over BP. Estimate produced using `https://github.com/mlco2/codecarbon`. Dashed line indicates the moment when performance is equal to maximal BP test accuracy.

Figure 6a shows how much time is needed to reach a specific percentage of the maximal test accuracy obtained using DBP, measured in terms of the percentage of training time needed to reach maximal accuracy for BP. Maximal BP test accuracy of 54.1% is reached by BP after 3.7 hours and by DBP after 1.5 hours, which is a 59% reduction in training time to achieve the same test accuracy. Maximal DBP test accuracy of 55.2% is reached by DBP after 2.3 hours, which is still a 38% reduction in training time.

Figure 6b provides a measurement of the associated reduction in carbon emission if we run the model to achieve a specific percentage of the DBP test accuracy. The figure shows that a reduction of 640 grams of carbon emission is achieved if we train DBP until it reaches 100% of the BP test accuracy. If we run DBP until it reaches 100% of the DBP test accuracy, we still achieve a reduction of 418 grams of carbon emission.

## 4 Discussion

Increasing the efficiency of deep learning is of key importance if we are to reduce the carbon footprint of AI. In this paper, we have shown that decorrelated backpropagation provides a viable path towards more efficient deep learning. Results show that, by replacing BP with DBP, we can achieve higher test accuracy using about half the number of training epochs. Results also show that both decorrelation ($\kappa = 0$) and whitening ($\kappa = 0.5$) provide similar performance gains, with whitening yielding slightly better results. This does not come at the expense of generalization performance, as shown in Fig. 11. Comparable results are obtained for other deep neural network architectures, as shown in Appendix G. This demonstrates that the performance gains are not architecture dependent. We further demonstrated a substantial reduction in carbon emission when using DBP over BP when training deep neural networks. Global carbon emissions can therefore be substantially reduced when applying DBP at scale.

The results also indicate that performance is maximized for some optimal value of $\kappa$ which most effectively balances the covariance and variance constraints. This optimal value is task-dependent. Appendix H shows that training a medium-sized neural network on the CIFAR10 task yields similar performance gains but now the optimal performance is obtained for $\kappa = 0$.

It should be mentioned that network-wide decorrelation comes with a number of additional advantages, which we did not further explore here. As shown in (Ahmad et al., 2023), decorrelated inputs allow for rapid linear approximation of non-linear functions, allowing for filter visualisation in deep networks with applications in explainable AI (Rudin, 2019; Ras et al., 2022). It also allows for network compression at virtually no computational overhead, which may further decrease carbon consumption at inference time (Wang, 2021). Intriguingly, input decorrelation and whitening have also been shown to be a feature of neural processing (Franke et al., 2017; Bell & Sejnowski, 1997; Pitkow & Meister, 2012; Segal et al., 2015; Dodds et al., 2019; Graham et al., 2006; King et al., 2013). The work presented here suggests that decorrelation may be an important factor when considering synaptic plasticity mechanisms.

A number of considerations do need to be taken into account when using DBP. First, DBP requires updating and storing of decorrelation matrices needed for input decorrelation. Hence, the algorithm introduces some computational overhead and increased memory requirements. This is reflected by the increased wall-clock time per epoch. To mitigate this, more effective use could be made of the sparseness structure of the decorrelation matrix. For instance, rather than learning a full matrix $\mathbf{R}$ we may choose to learn a lower-triangular matrix, which can be achieved by using the strictly lower triangular part of $\mathbf{C}$ in Eq. 7. Other approaches to reduce wall-clock time may be the use of low-rank approximations of $\mathbf{R}$, using higher layer-specific decorrelation learning rates, or only incorporating decorrelation in that subset of the layers which have the biggest impact in terms of reducing convergence time (Huangi et al., 2018).

Second, DBP needs proper fine-tuning since the stability of the algorithm depends on the two learning rates $\eta$ and $\epsilon$, whose ratio needs to be set appropriately such that the decorrelation rate throughout all network layers remains optimal. We empirically observe that this fine-tuning depends on the employed dataset, network architecture and loss function. Additional work on the interaction between these factors may provide insights into how to optimally set these parameters.

An important consideration here is scaling the updates according to layer size. Note that Eq. 7 ignores the layer size in the normalisation, which implies that decorrelation updates have a larger impact in larger layers

whereas whitening induced by $\kappa > 0$ has a smaller impact in larger layers. In simulation work, we observe that such normalisation ensures that the impact of decorrelation updates is comparable for different layer sizes. However, in our empirical work using very deep neural networks, we find that ignoring this normalisation actually improves performance. There are different reasons why this may be the case. First, the resulting stronger decorrelation in deep large layers when ignoring normalisation may have a beneficial impact since deeper layers are more strongly affected by multiple preceding nonlinearities. Second, the decorrelation loss that is minimized by Eq. 7 may not be the optimal metric to ensure fastest convergence. That is, there may be a discrepancy between the decorrelation gradient direction and the optimal gradient direction to align the inputs.

In this work, we demonstrated a significant speedup in deep residual networks, which are state-of-the-art models for computer vision. In our follow-up work, we aim to explore the efficiency gains that can be obtained for other computational tasks and network architectures. Exciting prospects are the use of decorrelated backpropagation for deep reinforcement learning (Mnih et al., 2015; Schrittwieser et al., 2020) or training of foundation models (Bommasani et al., 2021), both of which are notoriously resource intensive. We further expect that additional theoretical work on the optimal alignment of layer inputs combined with efficient low-level implementations will yield even larger gains in convergence speed. Concluding, by demonstrating the efficiency of decorrelated backpropagation in modern deep neural networks, we hope to contribute to reducing the energy consumption of modern AI systems.

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

## A    Comparison with zero-phase component analysis

Zero-phase component analysis (ZCA) (Bell & Sejnowski, 1995; 1997) is defined by the whitening transform $\mathbf{x} = \mathbf{C}\mathbf{x}_c$, where $\mathbf{x}_c = \bar{\mathbf{x}} - \boldsymbol{\mu}$ with $\boldsymbol{\mu}$ the mean over datapoints and $\mathbf{C}$ is the whitening transform. Let $\mathbf{X}$ be the $n \times d$ matrix consisting of $n$ $d$-dimensional centered inputs. ZCA uses

$$\mathbf{C} = \boldsymbol{\Lambda}^{-1/2}\mathbf{U}^{\top} \tag{8}$$

with $\mathbf{U}\boldsymbol{\Lambda}\mathbf{U}^{\top}$ the eigendecomposition of the covariance matrix $\boldsymbol{\Sigma} = \mathbf{X}\mathbf{X}^{\top}/n$.

Desjardins et al. (2015) propose to periodically (during training) measure the correlation, $\mathbf{C}$, at every layer of a deep neural network and to then apply eigendecomposition method to compute the ZCA transform. This approach to achieving an exact whitening transform is an alternative to what we attempt in this work but has a number of drawbacks. First, the complexity of computing the eigendecomposition itself has a cost equivalent to a matrix multiplication - meaning that this step alone is equally expensive to our decorrelation method. Second, atop this decomposition, Desjardins et al. (2015) further compute a matrix inverse of this transform which allows them to periodically compute an exact ZCA-based decorrelation within each layer of their network while also undoing its impact on the layer's computation (by modifying the layer's weight matrix $\mathbf{W}$) to ensure that the newly introduced transform does not change the network's output. We avoid the second step by simultaneously optimising for both the decorrelation and task at every layer and at every update step, without directly computing the ZCA transform. This provides the benefit of not requiring any matrix inversion computation and also ensures that we are continuously optimising the decorrelation transform rather than doing so at fixed intervals.

## B    Impact of sampling rate and frequency of decorrelation update

Figure 7 shows test performance as a function of wall-clock time for AlexNet trained on ImageNet at different sampling frequencies. Figure 8 shows performance at different sampling rates. It is shown that sampling 10 per cent of the batch appears sufficient for robust decorrelation while keeping computational overhead limited. Trying to save even more computation time by only updating the decorrelation matrix every 5 or 10 batches does not appear to be speed up convergence in this experiment.

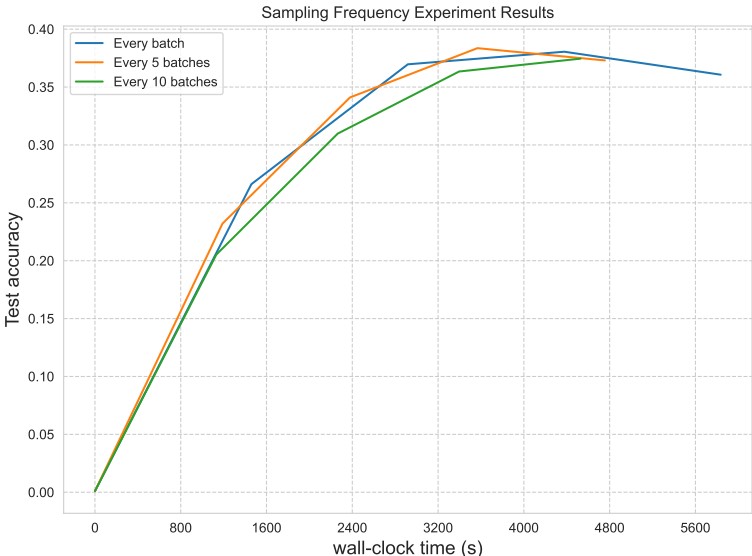

Figure 7: Test performance for AlexNet trained on ImageNet, sampling the decorrelation update every batch, every 5 batches and every 10 batches.

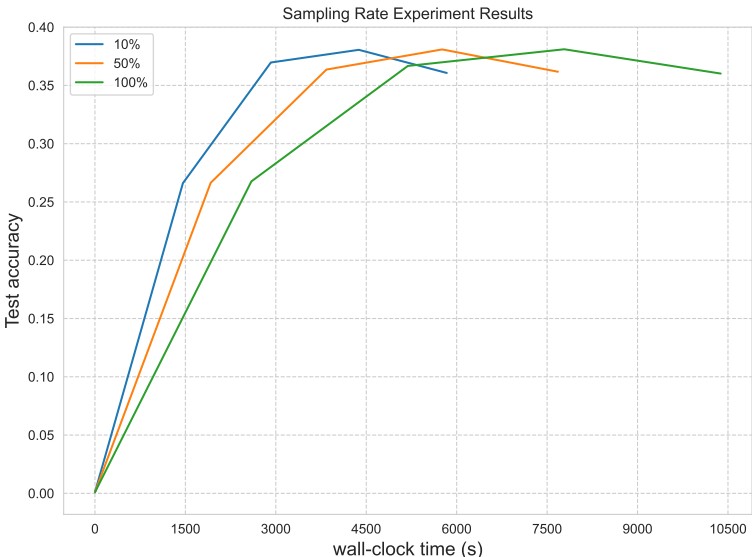

Figure 8: Test performance for AlexNet trained on ImageNet, sampling the decorrelation update from 100 per cent, 50 per cent and 10 per cent of the batch.

## C   Hyper-parameter optimization

A grid search across learning rate hyper-parameters was performed to ensure that each algorithm reached its optimal performance. An exploratory analysis revealed that although $3.2 \cdot 10^{-4}$ gave slightly better performance after five epochs for BP and DBP, later in training a learning rate of $1.6 \cdot 10^{-4}$ was better. Therefore, the latter was chosen for all algorithms, which had the convenient side effect of using the same learning rate for all algorithms, making comparison more straightforward. Figure 9 depicts the grid search outcomes.

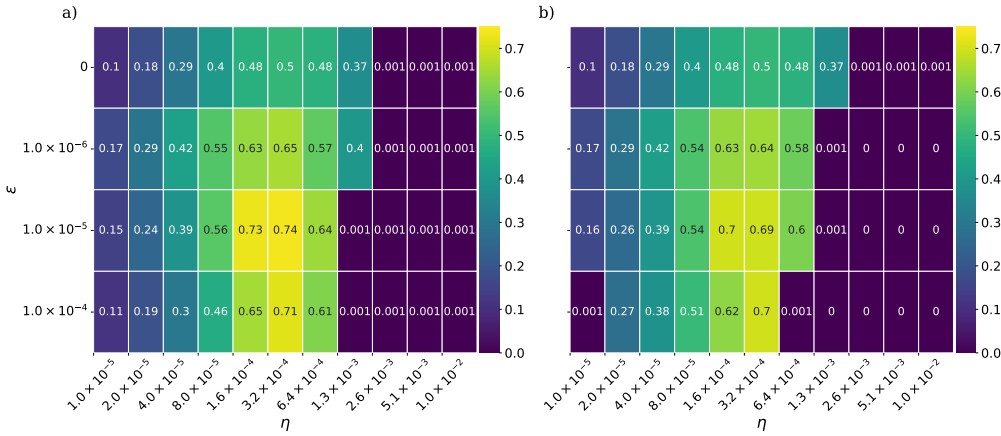

Figure 9: Train accuracy after five epochs for DBP when varying the BP learning rate $\eta$ and decorrelation learning rate $\epsilon$. Note that a decorrelation learning rate of zero corresponds to regular BP. a) Results using decorrelation ($\kappa = 0$). b) Results using whitening ($\kappa = 0.5$).

## D    Loss curves

Figure 10 reports the loss curves as a function of epochs and wall-clock time for the different algorithms, corresponding to the accuracies reported in the main text.

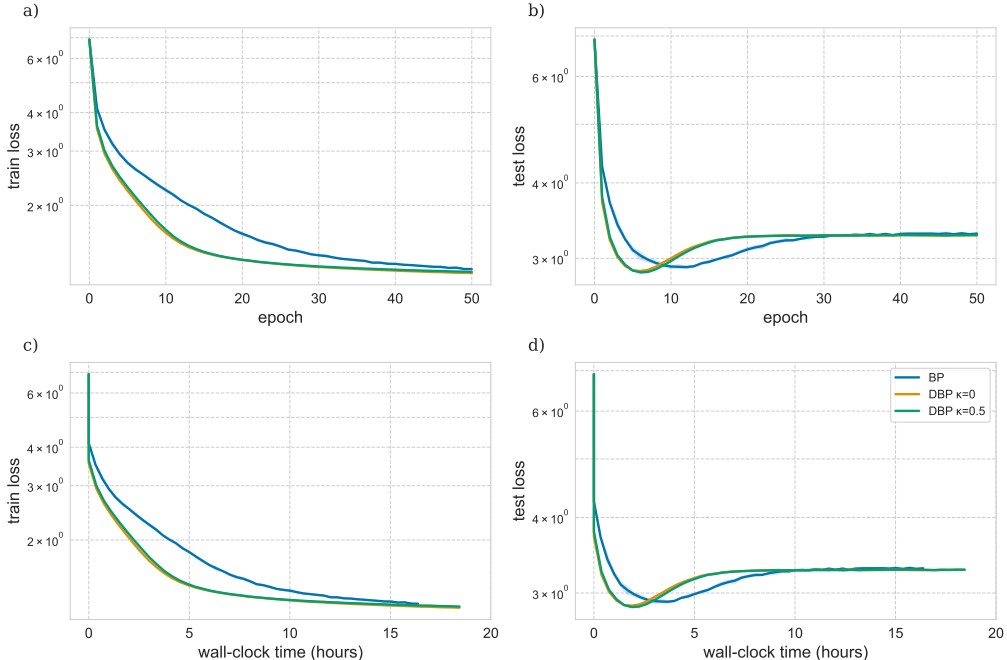

Figure 10: Performance of BP and DBP using $\kappa = 0$ (decorrelation) and $\kappa = 0.5$ (whitening). Reported results are the average of three randomly initialized networks, where minimal and maximal value are indicated by the error bars. a) Train loss as a function of the number of epochs. b) Test loss as a function of the number of epochs. c) Train loss as a function of wall-clock time. d) Test loss as a function of wall-clock time.

## E    Train versus test loss comparison

Figure 11 shows the train versus test losses for a ResNet-18 model trained on ImageNet for both BP and DBP. As DBP's test loss is slightly lower than the BP test loss at the same value of the train loss, we conclude that DBP yields slightly better generalization performance compared to BP.

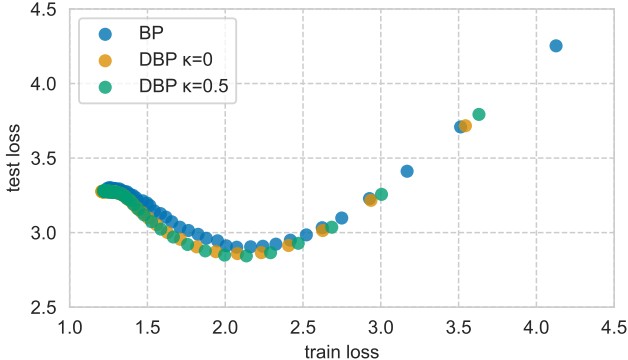

Figure 11: Scatterplot of train versus test loss for BP and DBP.

## F   Peak performance comparison

Table 1 provides an overview of peak performance for the different algorithms, as well as the epoch number and time at which peak test performance was achieved.

Table 1: Peak performance for BP, DBP in terms of loss and accuracy. Epoch and Time indicate the epoch number and wall-clock time at which peak accuracy was achieved. Best results shown in boldface, where we focus on test results as this quantifies generalization performance.

| Method | $\kappa$ | Loss Train | Loss Test | Accuracy Train | Accuracy Test | Epoch Train | Epoch Test | Time (hours) Train | Time (hours) Test |
|---|---|---|---|---|---|---|---|---|---|
| BP | - | 1.24 | 2.90 | 99.4 | 54.1 | 49 | 12 | 16.0 | 3.7 |
| DBP | 0 | 1.21 | 2.86 | 99.5 | 54.8 | 50 | **7** | 18.4 | **2.3** |
| DBP | 0.5 | 1.22 | **2.84** | 99.5 | **55.2** | 50 | **7** | 18.4 | **2.3** |

## G   Performance gains for other architectures

Figures 12 and 13 show performance of BP and DBP on ImageNet using AlexNet, ResNet18, ResNet34 and ResNet50 achitectures. Figures show accuracy as a function of epoch and wall-clock time, respectively. Results indicate that DBP's outperformance is present in all architectures, though the outperformance seems to decrease slightly as architectures get larger, especially in wall-clock time. The full set of reasons for this is yet unclear, though we did find that in larger architectures the computational overhead of decorrelation becomes greater, causing outperformance as a function of time to be diminished.

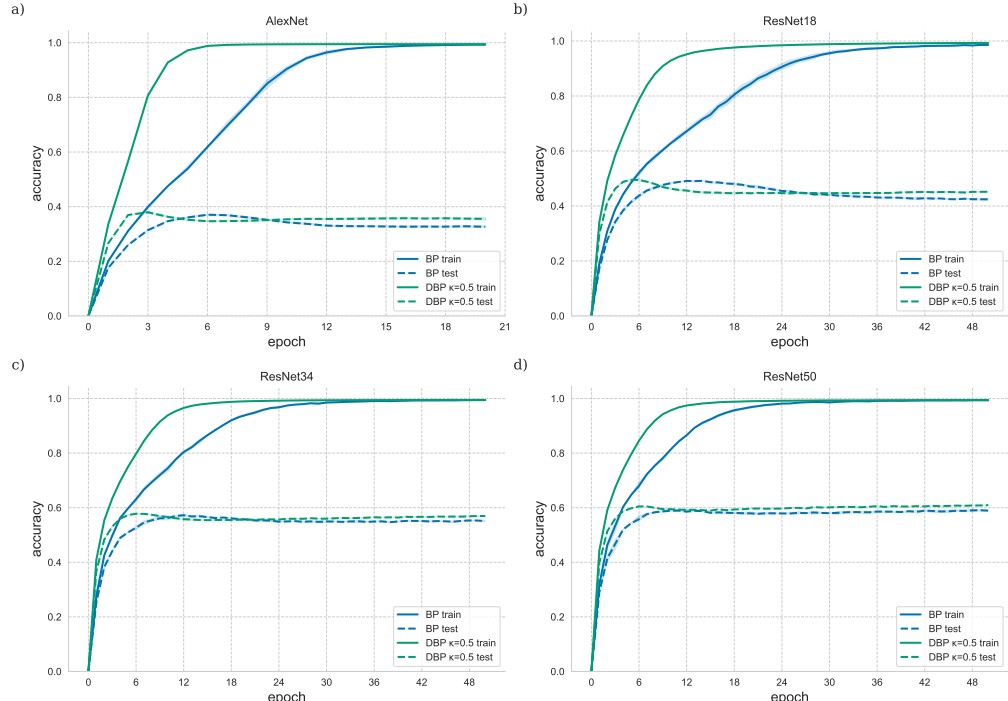

Figure 12: Train and test performance of BP and DBP on Imagenet as a function of epochs for different deep neural network architectures. a) AlexNet. b) ResNet18. c) ResNet34. d) ResNet50.

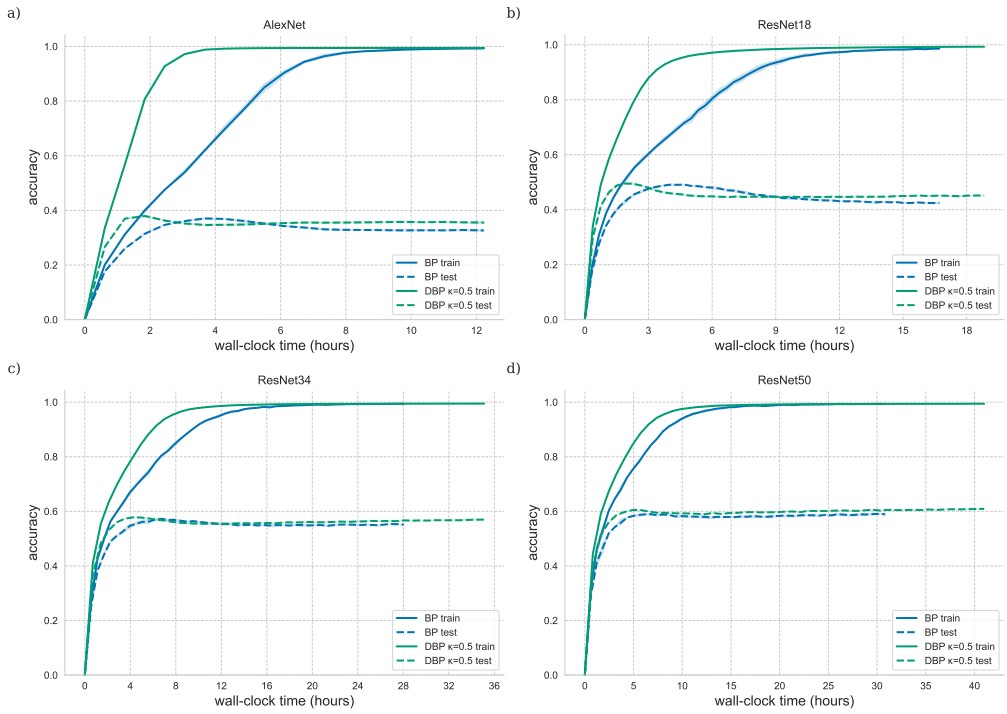

Figure 13: Train and test performance of BP and DBP on Imagenet as a function of wall-clock time for different deep neural network architectures. a) AlexNet. b) ResNet18. c) ResNet34. d) ResNet50.

## H   Impact of $\kappa$ parameter on CIFAR10 performance

Figure 14 shows train and test performance for a small three-layer ConvNet trained on CIFAR10 for BP and for several settings of DBP where the $\kappa$ parameter was varied. Setting $\kappa = 0$ reached the highest peak test accuracy for this particular task. Performance gains for this smaller model are even more pronounced yet result in a smaller reduction in carbon consumption due to the model's smaller size. Note further that peak test performance for DBP is larger than that of BP, again indicating improved generalization performance. A batch size of 256 and learning rate of $1 \times 10^{-3}$ were used in all experiments. The decorrelation/whitening learning rate was also set to $1 \times 10^{-3}$.

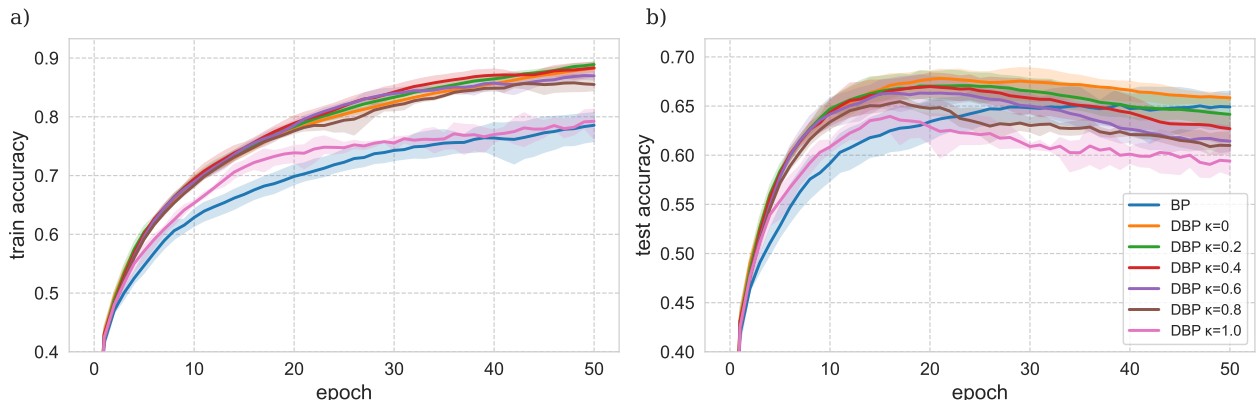

Figure 14: Performance on CIFAR10 for BP and several $\kappa$ settings of DBP. a) Train accuracy. b) Test accuracy.

