# OpenReview forum: "Efficient Deep Learning with Decorrelated Backpropagation"
_TMLR — Rejected by TMLR_

### Review · Reviewer_Gj5L · 2024-07-11

**Summary Of Contributions:**

This paper introduces a novel algorithm which induces network-wide input decorrelation using minimal computational overhead to speedup backpropagation in the neural network training.

**Audience:**

Yes

**Claims And Evidence:**

Yes

**Requested Changes:**

I do not have many concerns regarding this paper. Please demonstrate that the scaling rule is not only applicable to 18 layers.

**Strengths And Weaknesses:**

Strengths:

+ This is the first work showing that much more efficient training of very deep neural networks using decorrelated backpropagation is feasible.

+ The paper is well-written and all algorithmic details are clearly described.

+ The dataset is large enough to show the capability of the decorrelated backpropagation method.

Weakness:

- It is unclear why only a 18-layer ResNet is trained to evidence the algorithm efficiency. It seems that all results are only valid and tuned for 18 layers. Does the scaling rule also applicable to 10, 17, 20, or 50 layers?

---

> ### Author Response · Authors · 2024-09-04
> **Response to the reviewer's feedback**
>
> We thank the reviewer for their careful consideration of our work and for their constructive feedback.
>
> In response to this feedback, we have added several other architectures to our experiment set: AlexNet, ResNet34 and ResNet50. We see that results generally hold up, with slightly better performance gains for the smaller networks than the larger ones. See appendix F in the revised version of the paper.

---

### Review · Reviewer_vm6o · 2024-07-15

**Summary Of Contributions:**

The paper shows for the first time that efficient training of very deep NNs can be achieved via decorrelated backprobagation. The solution relies on a previously proposed method for network-wide decorrelation. Via a combination of these solutions a 2x improvement without loss of accuracy (or improvement). Experimental results support the general claims in the paper

**Audience:**

Yes

**Claims And Evidence:**

Yes

**Requested Changes:**

Mainly check the negatives. The paper is lacking in terms of robust evaluation to ensure the ideas/contributions hold more widely

**Strengths And Weaknesses:**

Positives
- Timely and important problem in the area as a solution good substantially improve training deep NNs
- Notation and writing is clear to help readers understand the concepts
- Experimental results support the claims

Negatives
- Technical depth is not that high. Mainly a combination of existing solutions it seems.
- No other baselines are used other than BP. This seems problematic
- Only a single architecture was used (if I understood correctly). Much more solutions should be tested to ensure the generalizability of results across architectures
- Much more datasets/applications/scenarios should be evaluated as well

---

> ### Author Response · Authors · 2024-09-04
> **Response to reviewer's feedback**
>
> We thank the reviewer for their careful consideration of our work and for their constructive feedback.
>
> - Concerning technical depth, please note that the paper provides a novel formulation of the decorrelation approach as well as specific technical developments (e.g. efficient application to convolutional layers) that were needed to afford scaling up towards large scale convolutional networks.
>
> - Regarding baselines other than BP. We decided that BP should be the baseline for this work, as it is the state of the art algorithm for networks like ResNet as used throughout the field. Other baseline algorithms will, in our experience, simply perform worse than BP. Note that for BP we optimize over hyperparameters to ensure that the best performance possible is obtained for BP.
>
> - Regarding experiments on other architectures. We appreciate that this is a valid point, therefore we have added experiments for ResNet34 and ResNet50 on the larger end and an experiment for AlexNet on the smaller end, to place our ResNet18 results in a broader context. The results hold up in the extra architectures, though a slight trend of decreasing performance gains can be seen as a function of network depth. The reason for this trend, as well as application of the method to completely different architectures, we regard as promising topics of further research. See appendix F in the revised version of the paper.

---

### Review · Reviewer_WXHd · 2024-09-12

**Summary Of Contributions:**

This paper introduces a novel decorrelated backpropagation (DBP) algorithm that enhances the efficiency of training deep neural networks (DNNs). The traditional backpropagation algorithm, while successful, has high computational costs and a significant carbon footprint. The authors demonstrate that input decorrelation can accelerate DNN training, but its implementation has been challenging due to computational overhead and stability issues. The DBP algorithm presented in this paper overcomes these challenges by using an efficient iterative local learning rule that decorrelates layer inputs across the network with minimal computational overhead. When applied to an 18-layer deep residual network, DBP achieves a two-fold speed-up in training time and higher test accuracy compared to standard backpropagation.

**Audience:**

Yes

**Broader Impact Concerns:**

No.

**Claims And Evidence:**

No

**Requested Changes:**

The application of the proposed method is not general and easy, many efforts need to be paid to explore the hyperparameters.
In contrast to stanard BP algorithm, the improvement of this method is not remarkable.

**Strengths And Weaknesses:**

Strengths:

1. The method avoids computing a large decorrelation matrix for the entire input, which would be computationally expensive, especially with large feature maps. By decorrelating local image patches instead, the method significantly reduces computational complexity. The approach is scalable as it adapts to the size of the input data. It uses a dimensionality reduction strategy that allows for efficient processing of image patches, making it suitable for deep learning applications with varying input sizes.
2. By applying a 1x1 convolutional kernel after local decorrelation, the method ensures more efficient learning of kernel weights, which can lead to better model performance. The method reduces the computational overhead during training by sampling only a small percentage of the batch for updating the decorrelation matrix, as opposed to using the entire mini-batch.
3. After training, only the condensed matrices A need to be stored, which further reduces storage requirements compared to storing the decorrelation matrix R and the weight matrix W separately. Combining the matrices W and R into a single matrix A and performing a single multiplication operation reduces the time required for the forward pass, leading to faster model inference.

Weaknesses:
1.  The method's performance might be sensitive to the choice of hyperparameters, such as the size of local image patches and the sampling rate for updating R. As a result, the generalization of this method needs to be verified.Since the method focuses on local decorrelation, there might be concerns about how well the model generalizes to new, unseen data, especially if the decorrelation is too aggressive.
2. On resnet18 and other networks in Appendix, the test accuracy is far below the official accuracies with standard BP method.

---

> ### Author Response · Authors · 2024-09-26
> **Response to reviewer's feedback**
>
> We thank the reviewer for their careful consideration of our work and for their constructive feedback.
>
> Regarding the size of the performance improvement, we have shown a higher maximum test accuracy of 55.2% for DBP versus 54.1% for BP, while taking only 2.3 hours to train versus BP’s 3.7 hours. The improvements are present in several deep architectures (see Appendix G).  Given the high overall carbon footprint of AI training, and its likely increase in the near future, we would consider this a significant contribution. The fact that the BP baseline is not state of the art is to be expected, because we compared the algorithms on a plain classification task, without data augmentation, complex learning rate schedules etc. in order to show an unconfounded comparison.
>
> Regarding the method’s sensitivity to hyper parameters, indeed the optimal settings for the additional hyper parameters relating to decorrelation need to be determined before training. The fact that we can downsample the input to about 10% when training decorrelation matrix R, was an empirical finding. We have added an Appendix B in which we explore several other sampling rates as well as sampling frequencies for R, to show the method’s sensitivity to this hyper parameter more clearly. The sensitivity to the decorrelation learning rate is explored in Appendix C.

---

### Decision · Action_Editor_5WeW · 2025-02-25

**Recommendation:** Reject

**Comment:**

Reviewers vm6o and Gj5L note the submission's clear writing. Having read the paper carefully, I agree as well.

In terms of interest to at least some individuals in TMLR's audience, the submission meets the bar for acceptance (see Audience section). There are however several concerns over the claims and evidence (see Claims And Evidence section) and as a result the submission does not meet the bar for acceptance at TMLR in its current form.

**Audience:**

Reviewers generally agree that the submission is of interest to at least some individuals in TMLR's audience. Reviewer vm6o finds the problem "timely and important", and in their official recommendation Reviewer WHXd notes that the paper presents good theory and would be interesting to researchers.

Reviewer vm6o is concerned with the submission's lack of technical depth. In their response, the authors highlight the paper's contributions in terms of its efficient application to convolutional layers. From TMLR's standpoint, "technical depth" is not in itself a criterion by which a paper should be accepted or rejected and this concern does not influence the final decision.

There is a sense in which the submission would be appealing to a much broader audience if it made the demonstration that DBP can be successfully applied to modern architecture families and models like e.g. diffusion models and Transformer-based architectures. Overall, though, the submission meets the bar for the Audience criterion.

**Claims And Evidence:**

The submission's claims are fairly strong and would require more substantial evidence to back them up:

*  "Here, we show for the first time that much more efficient training of very deep neural networks using decorrelated backpropagation is feasible."
* "This demonstrates that decorrelation provides exciting prospects for efficient deep learning at scale."
* "In this paper, we show that deep learning can be made much more efficient by enforcing decorrelated inputs throughout the network."
* "In the following, we show that DBP yields a two-fold reduction in training time, while achieving better performance compared to regular BP. Hence, widespread application of our approach can yield a substantial reduction in the carbon consumption of modern deep learning." [Note: the abstract qualifies that the two-fold reduction pertains to an 18-layer deep residual network, but the main text does not.]
* "In this paper, we have shown that decorrelated backpropagation provides a viable path towards more efficient deep learning. Results show that, by replacing BP with DBP, we can achieve higher test accuracy using about half the number of training epochs."
* "Comparable results are obtained for other deep neural network architectures, as shown in Appendix G. This demonstrates that the performance gains are not architecture dependent."
* "We further demonstrated a substantial reduction in carbon emission when using DBP over BP when training deep neural networks. Global carbon emissions can therefore be substantially reduced when applying DBP at scale."
* "In this work, we demonstrated a significant speedup in deep residual networks, which are state-of-the-art models for computer vision."

Taken together, those claims give the impression that the proposed approach, if universally adopted, would immediately have a substantial impact on deep learning efficiency and (as a result) on carbon emissions of deep learning. Reviewers vm6o's and Gj5L's concern over the fact that results are only presented for ResNet-18 are indicative of this. While the authors' addition of experiments for AlexNet and various flavors of ResNet addresses this concern to some extent (Reviewer Gj5L expresses that this resolves their concern in their official recommendation), image classification only represents a subset of modern deep learning applications, and the evidence presented does not convincingly establish that DBP would provide similar benefits in Vision Transformer-based classifiers, diffusion models, or large language models, to name a few. To be clear, the issue here is not that the bar for evidence is to demonstrate benefits on all those models and architectures and the evidence does not meet that bar; rather, the issue is that the claims made in the submission make it so that this is the evidence that would be needed to back up those claims.

There are two possible ways to address this shortcoming: either expand the scope of the experiments to more problem settings and architecture families, or restrict the scope of the claims to the image classification setting and convolutional network architectures.

Regarding the added results on AlexNet and various flavors of ResNet, the authors mention a "slight trend of decreasing performance gains [...] as a function of network depth". Given the claims that "[the paper] demonstrates that decorrelation provides exciting prospects for efficient deep learning at scale" and "global carbon emissions can therefore be substantially reduced when applying DBP at scale", and given the heavy trend towards larger and larger model architectures, this warrants further discussion and exploration. Specifically, do the observed diminishing returns eventually close the gap between DBP and BP at very large model sizes, or is there still a significant benefit in terms of training efficiency and carbon emissions?

While reading the paper carefully, I also noticed a methodological issue that was not brought up by reviewers. The claims made about DBP's efficiency compared to BP hinge upon the epoch or time to the best test accuracy observed, but these claims can only be made with the benefit of hindsight. If we are looking at the test accuracy to decide at what point in training to compare the two approaches, then the test accuracy is no longer an unbiased estimate of the risk and claims like "DBP's test accuracy (55.2%) also peaks above BP's test accuracy (54.1%)" are no longer meaningful. This is a problem that comes up in the experiments presented in Sections 3.2, 3.3, and 3.4. To be completely rigorous, the experiments should make a stopping decision based on the validation loss or accuracy, and only then look at the test accuracy.

**Resubmission Of Major Revision:**

The authors may consider submitting a major revision at a later time.